# Implication of Central Nervous System Barrier Impairment in Amyotrophic Lateral Sclerosis: Gender-Related Difference in Patients

**DOI:** 10.3390/ijms241311196

**Published:** 2023-07-07

**Authors:** Hugo Alarcan, Patrick Vourc’h, Lise Berton, Isabelle Benz-De Bretagne, Eric Piver, Christian R. Andres, Philippe Corcia, Charlotte Veyrat-Durebex, Hélène Blasco

**Affiliations:** 1Laboratoire de Biochimie et Biologie Moléculaire, CHRU Bretonneau, 2 Boulevard Tonnellé, 37000 Tours, France; patrick.vourch@univ-tours.fr (P.V.); isabelle.benz-debretagne@univ-tours.fr (I.B.-D.B.); piver_e@univ-tours.fr (E.P.); christian.andres@univ-tours.fr (C.R.A.); charlotte.veyratdurebex@univ-tours.fr (C.V.-D.); helene.blasco@univ-tours.fr (H.B.); 2UMR 1253 iBrain, Université de Tours, Inserm, 10 Boulevard Tonnellé, 37000 Tours, France; philippe.corcia@univ-tours.fr; 3Service de Neurologie, CHRU Bretonneau, 2 Boulevard Tonnellé, 37000 Tours, France

**Keywords:** amyotrophic lateral sclerosis, QAlb, central nervous system barrier, C9orf72, blood–brain barrier, spinal cord barrier

## Abstract

Central nervous system (CNS) barrier impairment has been reported in amyotrophic lateral sclerosis (ALS), highlighting its potential significance in the disease. In this context, we aim to shed light on its involvement in the disease, by determining albumin quotient (QAlb) at the time of diagnosis of ALS in a large cohort of patients. Patients from the university hospital of Tours (*n* = 307) were included in this monocentric, retrospective study. In total, 92 patients (30%) had elevated QAlb levels. This percentage was higher in males (43%) than in females (15%). Interestingly, QAlb was not associated with age of onset, age at sampling or diagnostic delay. However, we found an association with ALS functional rating scale-revised (ALSFRS-r) at diagnosis but this was significant only in males. The QAlb levels were not linked to the presence of a pathogenic mutation. Finally, we performed a multivariate survival analysis and found that QAlb was significantly associated with survival in male patients (HR = 2.3, 95% CI = 1.2–4.3, *p* = 0.009). A longitudinal evaluation of markers of barrier impairment, in combination with inflammatory biomarkers, could give insight into the involvement of CNS barrier impairment in the pathogenesis of the disease. The gender difference might guide the development of new drugs and help personalise the treatment of ALS.

## 1. Introduction

Amyotrophic lateral sclerosis (ALS), characterised by the degeneration of both upper and lower motor neurons, is the most common motor neuron disorder with adult onset [1]. This is a devastating disease, as most patients die from respiratory failure after 3 to 5 years after the onset of symptoms [1]. In most cases, the cause of the disease remains unknown, but more than 30 genes are known to cause the disease [2]. The most common mutations responsible for inherited ALS are present in four genes named *C9orf72*, *TARDBP*, *SOD1* and *FUS*. Several mechanisms have been associated with the disease, including the aggregation and accumulation of ubiquitinated protein inclusions in motoneurons, alterations in mRNA processing, glutamate-mediated excitotoxicity, oxidative stress, mitochondrial dysfunction or neuroinflammation [3,4,5,6,7,8]. However, numerous drugs targeting main pathological mechanisms involved in ALS have been tested in clinical trials, but failed to demonstrate a significant benefit, highlighting that the pathogenesis of ALS remains partially understood. Impairment of central nervous system (CNS) barriers may contribute to its mysterious pathogenesis and probably impacts the development and administration of therapeutic candidates [9].

The blood–brain barrier (BBB) and blood–spinal cord barrier (BSCB) have a complex composition, including some entities forming the neurovascular unit (NVU) [10]. They protect the CNS from neurotoxic compounds or drugs representing a major limit to the development of therapeutics targeting the brain. BBB/BSCB impairment has been known in ALS for a decade [11,12,13] and has been discussed in recent reviews highlighting its significance in the disease, via its putative implication in the pathogenesis mechanism, as a biomarker of disease outcome, and finally as required drug development strategy parameter [9,14,15]. Unfortunately, data published on BBB/BSCB disruption are too few to conclude on its role in the course of the disease. Moreover, there is a lack of robust and non-invasive markers to evaluate the integrity or functionality of the CNS barrier routinely available in living patients [16]. Despite its limitations, the albumin quotient (QAlb) is the most commonly used marker for reflecting BBB/BSCB permeability. Elevation in QAlb levels has been reported in 20–50% of ALS patients [17,18,19] and its association with the progression of the disease remains controversial [18,20,21,22]. Moreover, BBB/BSCB impairments have especially been studied in the context of *SOD1* mutated models of the disease [9]. These studies reported a leakage to blood-derived (haemoglobin) or injected molecules (e.g., Blue Evans, dextrans etc.) and an impairment of the ultrastructure of the CNS barriers or CNS barriers components (e.g., loss of pericyte coverage, alteration of the basement membrane) in a consensual way [12,13,23]. However, the onset of these barrier alterations and their pathophysiological mechanism remain unclear. Overall, knowledge about the contribution of other genetic forms (than *SOD1*) to these alterations is lacking.

In this context, we aim to shed light on the involvement of CNS barrier impairment in the disease, with determination of QAlb and other cerebrospinal fluid (CSF) markers at the time of ALS diagnosis in a large cohort of patients. We also explored these parameters according to the presence or absence of an expansion in the *C9orf72* gene to give insights into the pathophysiological mechanism of the disease in this subgroup of ALS patients.

## 2. Results

### 2.1. Clinical and Demographic Characteristics

In total, 307 patients were included in this study. Demographic and clinical characteristics are shown in Table 1. Briefly, 46.6% of subjects were female and 71% had spinal onset. The median disease duration was 26 months. Information about death was missing for 33.2% of patients.

### 2.2. Sex-Related Difference of Biological Features

We observed a high correlation between the biological variables in the entire population (Figure 1A), especially between QAlb and Immunoglobulin G (IgG) quotient (QIgG), and this was also observed in male and female patients. In total, 92 patients (30%) had elevated QAlb according to the age-related upper reference limit. This percentage was higher in males (43%) than in females (15%) (Figure 1B). On the other hand, only one female and no males exhibited an elevation in IgG index (>0.7). As expected, we observed marked differences between males and females for QAlb, CSF protein, CSF IgG, CSF albumin and QIgG. Males had significantly higher levels than females for all parameters, as illustrated with QAlb (median in males = 0.76%, quartiles = 0.59–1.02%) (median in females = 0.58%, quartiles = 0.47–0.74%) (Figure 1C). According to the high correlation between variables, the sex-related differences and the absence of elevated IgG index, we decided to focus the rest of the investigation on QAlb and we analysed females and males separately. 

### 2.3. Association of QAlb with Clinical and Genetic Parameters

Correlation of QAlb with the clinical characteristics of the patients is shown in Figure 2. Briefly, QAlb was not associated with age of onset or age at sampling, either in males or females. Interestingly, it was also not associated with the diagnostic delay. We also observed an association with the site of onset, forced vital capacity (FVC) and weight loss from the onset of the disease. Interestingly, we found an association with ALS functional rating scale-revised (ALSFRS-r) and progression rate of ALSFRS-r (dALSFRS) at diagnosis but this was significant only in males.

Genetic status about *SOD1* and *C9orf72* was available for 247 patients. In total, 14 patients had a pathological hexanucleotide expansion (number of repetitions ≥ 30) and 4 patients had a mutation in *SOD1* gene. There was no difference in QAlb levels between patients with pathogenic gene mutation and those without, whether in males (*p* = 0.78) or females (*p* = 0.16) (Figure 3A). The exact number of repeats was available for 171 patients (clinical characteristics of this subset are available in Appendix A). We explored the association between QAlb levels and the three groups compiled according to the repeat number of the longest allele. We found that males having a number of repeats >2 and <30 had lower QAlb than males with 2 repeats (Figure 3B). However, the other comparisons were not significant.

### 2.4. Association between QAlb at Diagnosis and Disease Outcome

We evaluated the correlation between QAlb and variations in weight, FVC and ALSFRS-r score over 1 year, but we found no association, neither in male nor female patients (Figure 2). Kaplan–Meier curves according to the median value of QAlb in males and females displayed association with survival in males only (Figure 4).

Using the univariate Cox proportional-hazards model, age of onset, site of onset, FVC, ALSFRS-r, dALSFRS, variation in reference weight at diagnosis, diagnostic delay and QAlb had an adjusted *p*-value of <0.2 in males. As none of these variables presented high collinearity, they were all included in the multivariate model (Table 2. QAlb, dALSFRS-r, FVC, onset site and diagnostic delay remain independently associated with survival in ALS. Concerning females, onset site, ALSFRS-r, dALSFRS-r, FVC, variation in reference weight at diagnosis and diagnostic delay had a *p*-value <0.2 in the univariate Cox model and were included in the multivariate model (Table 2). Only diagnostic delay and FVC remained independently associated with survival.

## 3. Discussion

This retrospective study encompassing 307 ALS patients showed signs of altered BBB in 30% of ALS patients from the time of diagnosis. Moreover, a higher level of QAlb at diagnosis was associated with reduced survival in male patients. To our knowledge this study is also the first to investigate the correlation between QAlb and the number of hexanucleotide repetitions in *C9orf72*.

### 3.1. Signs of BBB/BSCB Alteration Are Inconsistent in ALS but Are Associated with Survival in Males

Despite QAlb not being a perfect reflection of BBB/BSCB integrity [9], it is currently the most commonly used marker in clinical practice. We found an elevation of QAlb in 43% and 15% of male and female patients, which is in agreement with previous studies reporting elevation of QAlb in 10–50% of ALS patients [17,18,19]. The results of these studies are weighted by the limited number of patients and the heterogeneity or blur about the timing of QAlb evaluation regarding diagnosis of the disease. A source of the variability reported might be the different upper reference limits of QAlb used [18,20,22]. Thus, BBB/BSCB disruption would not appear in all individuals, at least at the time of diagnosis, and this illustrates the complex heterogeneity of ALS and could suggest the involvement of different pathophysiological mechanisms between patients.

BBB/BSCB impairment has been reported in various conditions, such as neurodegenerative disorders, stroke, epilepsy, traumatic brain injury or diabetes [11,21]. In most of the cases, it is difficult to conclude as to the involvement of BBB/BSCB impairment as a cause or a consequence of the disease. This is also the case in ALS, in which various types of evidence for BBB/BSCB alterations, such as ultrastructure impairment, immune cell extravasation or transporter dysfunction have been reported in animal and human post-mortem studies [9]. Oxidative stress is recognised to be involved in ALS pathophysiology. Reactive oxygen species are also involved in CNS barrier disruption as they can activate matrix metalloproteinases (MMPs), modulate cadherins and tight junction (TJ) protein expression and distribution, activate the NFkB pathway (leading to inflammation) or create oxidative damage to cellular molecules [24]. This could contribute to alteration of the barriers observed in ALS, despite an elevation in ROS possibly also being a consequence of such alteration. Other causes of barrier damage include inflammatory cytokines and extravasation of immune cells [25]. Indeed, numerous cytokines (such as IL-1β, IL-6,IL-9-IL-17, TNF-α and CCL2) are involved in the reduction in TJ protein expression or incorrect localisation, especially Claudin-5 [26]. IL-9 might alter Claudin-5 expression by inducing the secretion of VEGF-A from astrocytes [26]. Furthermore, IL1-β would promote immune cells infiltration across the BBB by the upregulation of α5β1 integrin-dependent signalling [27]. Recently, Zamudio et al. reported a novel role of TDP-43 [28]. In their study, TDP-43 overexpression in wild-type mice could induced BBB alteration, as suggested by the infiltration of CD3+ and CD4+ T cells, Immunoglobulin G and the presence of cerebral microbleeds, which were associated with increased markers of endothelial cell activation: ICAM1, VCAM and caveolin). They also observed an elevation of neutrophil recruitment into the brain, associated with an increased expression of chemotactic molecules KC-GRO and MIP-1α. This infiltration of neutrophils was exacerbated in the presence of systemic inflammation. Supposing that peripheral inflammation is actually the cause of BBB/BSCB alteration observed in ALS, the heterogeneity of inflammatory response in ALS patients [29] might contribute to the inconsistency in observed BBB/BSCB leakage.

The association of BBB/BSCB alteration with the progression of the disease in living patients remains controversial as some studies reported shorter survival with elevation of QAlb [15,16], while others did not find any associations [13,17]. The lack of clarity about the timing of CSF collection complicates the interpretation of these findings. Accordingly, we limited these factors by including a large cohort of ALS patients (*n* = 307), considering only samples collected at the time of diagnosis and analysing males and females separately. To note, in our cohort, QAlb levels did not correlate with the diagnostic delay, thus suggesting that the alteration in BBB/BSCB integrity observed may be independent of ALS progression in the early stage of the disease. Thus, we found an association between QAlb with ALSFRS-r score at diagnostic and the survival of the disease, but only in male patients. BBB/BSCB alteration might contribute to disease progression via different mechanisms, including the entry of toxic substances, imbalance of ions and neurotransmitters or release of cytokines, leading to neuroinflammation and subsequent motoneuron degeneration [24]. Free iron resulting from red blood cell leakage and serum proteins such as plasminogen and fibrinogen can also contribute to motoneuron loss via the generation of ROS, degradation of laminin and activation of microglia [30]. The oncotic pressure of the plasma is primarily maintained by albumin. The leakage of albumin inside the CNS might be responsible for the extracellular oedema observed in ALS patients and animal models of the disease [9], obstructing the CNS blood flow, with hypoxia contributing to neurodegeneration [31].

### 3.2. A Gender Effect to Take into Consideration

There is a sex-related difference in QAlb, as reported by Parrado-Fernández et al. in more than 27,000 patients (diseases not mentioned) and 335 healthy controls, with similar differences between males and females in the two populations [32]. It supports the determination of new reference intervals considering sex, in addition to age. The mechanism of this sex-related difference remains to be explored, but it may involve a reduction in the CSF flow rate and a higher body mass index in males [32,33]. Even if oestradiol may regulate the expression of MMP [34], the persistence of this sex-related difference at menopause or puberty [32] does not support a central role of hormonal factors in this phenomenon. 

This difference is also found in ALS, as reported here (even if females are older in our cohort) and in other studies [16,17]. To note, in our cohort, QAlb levels were 1.3 fold higher in males, in keeping with the previous study [32]. Interestingly, QAlb did not correlate with age at sampling or age of onset in either males or females. This could reflect that an ALS-driven elevation of QAlb would overtake the age-related increase observed in healthy populations, as already discussed [22]. According to this gender difference, we believe it is crucial to stratify data by sex to understand the implication of BBB/BSCB alteration in ALS. For example, the association between QAlb and the site of onset has previously been reported with increased values and association with survival in patients presenting with spinal onset [20]. However, as presented in this study and elsewhere, male patients have a greater likelihood of onset in the spinal regions, while females tend to have onset in the bulbar region [35]. Oestrogen may protect the females from the harmful effects of the alteration of CNS barriers. Oestrogen has been reported to counter inflammation-induced changes in zonula occludens-1 and occludin and to reduce immune cell extravasation in the presence of an inflammatory stimuli [36]. The protective effect of oestrogen would be mediated by the anti-inflammatory protein annexin A1 (ANXA1), through an increase in its synthesis via ERβ and the modulation of its posttranslational modification via GPR30 [36]. We can also note that males appear to be more susceptible to oxidative stress than females, which could enhance the level of ROS production following free iron leakage [37]. Our results suggest the usefulness of deciphering in more detail this sex-related difference in BBB leakage. This difference might also guide the development of new drugs and help personalise the treatment of ALS. Actually, male and female patients may benefit from different therapeutic targets (such as restoring BBB/BSCB repair in males) and different strategies to deliver the therapeutic to its central target.

### 3.3. C9orf72 Hexanucleotide Expansions Are Not Associated with BBB/BSCB Leakage in ALS

Most of the studies investigating BBB/BSCB integrity in ALS have been conducted in animal models, and almost exclusively on *SOD1* mutated rodents. However, the most commonly known genetic cause of ALS is *C9orf72* hexanucleotide expansion [2]. Unfortunately, BBB/BSCB has been little explored in *C9orf72* mutated models or patients carrying a pathological expansion. The mechanism of P-glycoprotein (P-gp) upregulation observed in the spinal cord of ALS patients and *SOD1* mutated animal models has been studied, using human iPS-derived endothelial cells and astrocytes from different causal factors (*sporadic*, *SOD1-AV4* and *C9orf72*) [38]. Astrocytes are believed to modulate P-gp expression in endothelial cells via glutamate secretion. However, C9-astrocytes did not increase P-gp expression in C9-endothelial cells, suggesting a distinct mechanism of the increased P-gp. Recently, different transport processes across the BBB have been evaluated in early symptomatic transgenic *C9orf72* BAC (C9-BAC) mice [39]. The authors did not report a specific BBB alteration caused by *C9orf72* pathological expansions. Given the small number of patients in most studies evaluating QAlb in ALS, an association of BBB/BSCB alteration with a *C9orf72* genotype rarely been evaluated or reported. Here, we did not find significant differences in QAlb levels between inherited (*SOD1* or *C9orf72*) and idiopathic forms of ALS. We also searched for an association between QAlb levels and the number of hexanucleotide expansions because *C9orf72* intermediate repeats might cause specific phenotypes, as previously reported [40]. Thus, it does not appear that *C9orf72* expansions are likely to be more involved in BBB/BSCB alterations observed in ALS than other causal factors.

This study has some limitations. The main limit of the project is its retrospective design. However, the large size of the cohort provides us the opportunity to have a relevant scientific background, necessary to plan more ambitious studies. According to the free access to the data and not to the biobank, a hormonal measurement could not be performed. It would be interesting to progress the potential protective effect of oestrogen in females. In the same way, the measurement of inflammatory markers and other molecules reflecting CNS barriers status (neuron-specific enolase (NSE), S100) could not be performed. The genetic status of *TARDBP* and *FUS* or even other genes were not available for all patients as genetic screening evolved over time and these variables were not included in this study. A similar preliminary project including other genetic forms would be interesting as it could also improve our knowledge about the contribution of the main mutations/genes on CNS barrier alterations. A multicentre study must be planned to include more patients, better explore the relationship between genetic modification and barrier alterations and go deeper into the pathophysiological mechanism exploration via hormonal and inflammatory exploration.

## 4. Materials and Methods

### 4.1. Study Design

This is a monocentric, retrospective study that included patients with ALS, diagnosed at the ALS centre in Tours (France), between January 2008 and April 2022. Patients were informed in writing regarding the use of their clinical and biological data for research aims and given the right to refuse such uses; however, none refused. Moreover, all patients were informed about the data obtained and about their right to access these data, according to articles L.1121-1 and R1121-2 of the French Public Health Code (CNIL n°2019-071). All the analysed parameters were part of the clinical routine. All patients were diagnosed as “definitive”, “probable”, “probable with laboratory result support” or “possible” ALS by the revised El Escorial criteria to establish the diagnosis [41]. Information about the diagnosis, the date of onset, age of onset, sex, the site of onset, genetic status concerning *C9orf72* and *SOD1*, reference weight and also weight, forced vital capacity (FVC) and the ALSFRS-r score at the first consultation in our ALS centre and 12 months later was recorded. Patients older than 18 for whom the ALSFRS-r score, weight and also an assessment of QAlb within 90 days of the first consultation in our centre were available were enrolled. Respiratory onset cases (*n* = 6) were excluded. The time between the date of the first symptoms and the date of death was calculated to determine the duration of the disease. Disease outcome parameters were also defined as the percentage variation in weight, FVC and ALSFRS-r score over a year. The variation in reference weight was determined using the following formula: (Weight at consultation − Reference weight)/Reference weight × 100. dALSFRS-r was calculated as follows: (48 − ALSFRS-r)/disease progression (months).

### 4.2. Analyte Quantification

Serum and CSF samples were collected conjointly for routine biological exploration around the first consultation. The analytical methods were carried out in accordance with relevant guidelines. CSF protein, serum and CSF Albumin and IgG were determined for all patients using a COBAS 6000 analyser^®^ (Roche Diagnostics, Meylan, France) and immunoturbidimetric assay. QAlb and QIgG were calculated using the following formula: QAlb (%) = CSF albumin/serum albumin × 100 and QIgG (%) = CSF IgG/serum IgG × 100. To determine if QAlb was elevated, we compared the value to the age-related upper reference limit as previously reported: (4 + age/15) ×10^−1^ [42]. IgG index was determined as follow: Ig Index = (CSF IgG/serum IgG)/(CSF albumin/serum albumin).

### 4.3. Genetic Analysis

Informed consent for genetic analysis was obtained and genomic DNA was extracted from whole blood samples. The presence of a pathological hexanucleotide expansion GGGGCC (repeat number ≥ 30) in the *C9orf72* gene was determined using repeated primed polymerase chain reaction followed by fragment sizing on a 3130xl Genetic Analyser (Thermo Fisher Scientific, Waltham, MA, USA). Three groups were determined according to the number of *C9orf72* repeats on the longer allele: patients carrying 2 repeats, patients carrying >2 and <30 repeats, and patients carrying ≥30 repeats. Data were analysed using GeneMapper software (version 6). The five exons of *SOD1* were analysed using sanger sequencing on the same analyser. Patients were considered to have an inherited form of ALS if a pathogen variant was identified.

### 4.4. Statistical Analysis

A sex-related difference is known for QAlb values [32], so we compared clinical and biological characteristics between males and females in our cohort to plan an analysis within sex subgroups in a second phase. For these comparisons, Mann–Whitney and Chi-square test were used for continuous and categorical variables, respectively.

Then, Pearson’s correlation coefficient was used to evaluate the correlation between biological features and continuous clinical characteristics of ALS in male and female patients.

Kruskal–Wallis test was used to evaluate the difference in QAlb values between the three groups, according to the number of *C9orf72* expansions.

Finally, survival analysis was conducted. First, the univariate Cox proportional-hazards model was used to determine the hazard ratios (HRs) and the Confidence Intervals (CIs) of the biological and clinical variables (site of onset, FVC, ALSFRS-r, dALSFRS, weight, variation in reference weight at diagnosis, diagnostic delay) on survival. To determine the independent association of these variables with ALS survival, variables with a *p*-value < 0.2 in the univariate models were included in a multivariate Cox proportional-hazards model after removal of highly correlated variables (r > 0.8).

The *p*-values were adjusted for multiple comparisons by the False Discovery Rate (FDR) method. The *p*-values < 0.05 were considered statistically significant. Analyses were performed using RStudio version 2022.02.3.

## 5. Conclusions

To conclude, we found that BBB/BSCB integrity is altered in many male ALS patients and that the QAlb level is associated with survival in this subgroup but not in females. We did not report a specific association between BBB/BSCB alteration and inherited forms of the disease. A longitudinal evaluation of barrier impairment markers (QAlb or other biomarker candidates such as NSE and S100β) throughout the course of ALS, in combination with inflammatory biomarkers, could give insight into the involvement of BBB/BSCB alteration in the pathogenesis of the disease.

## Figures and Tables

**Figure 1 ijms-24-11196-f001:**
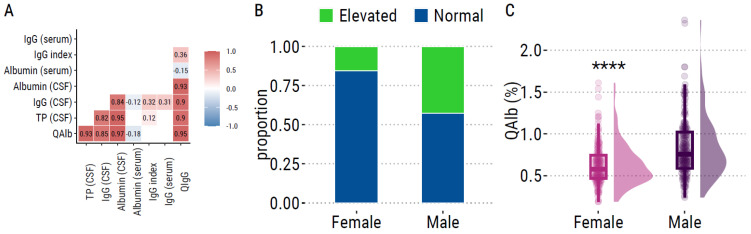
Association of QAlb with gender in ALS. (**A**) Correlation matrix between variables in all patients, using Pearson coefficient correlation. Only boxes with significant correlation are filled with coefficient value (r). White boxes represent non-significant correlation. Blue and red boxes represent significant negative and positive correlation, respectively. (**B**) Proportion of males and females with an alteration in BBB according to the age-limited reference value: (4 + age/15) × 10^−1^. (**C**) Raincloud plots of QAlb values between male and female ALS patients. ****: *p* ≤ 0.0001.

**Figure 2 ijms-24-11196-f002:**
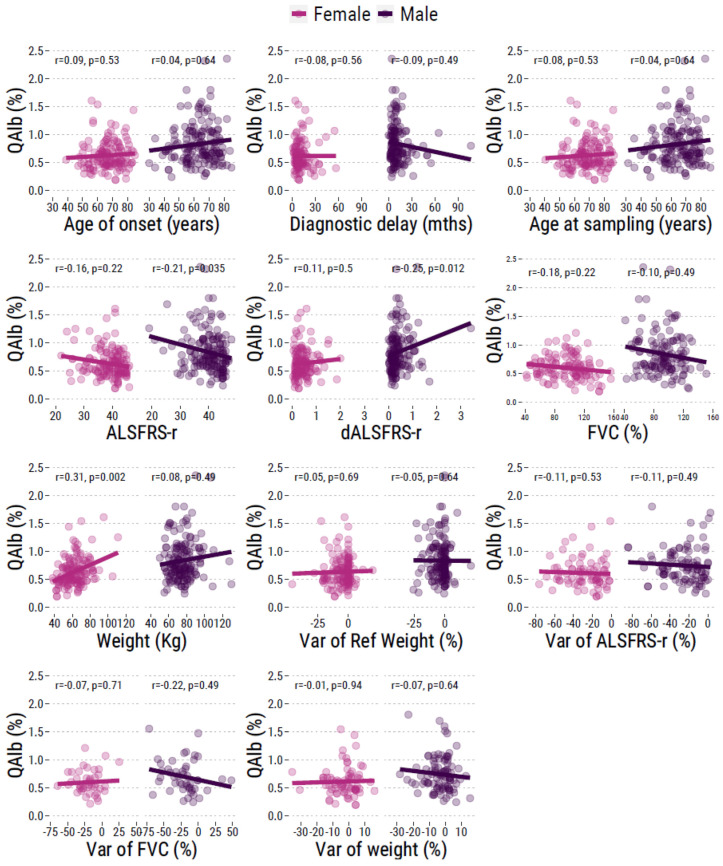
Correlation between QAlb value and clinical parameters of ALS, in male and female patients.

**Figure 3 ijms-24-11196-f003:**
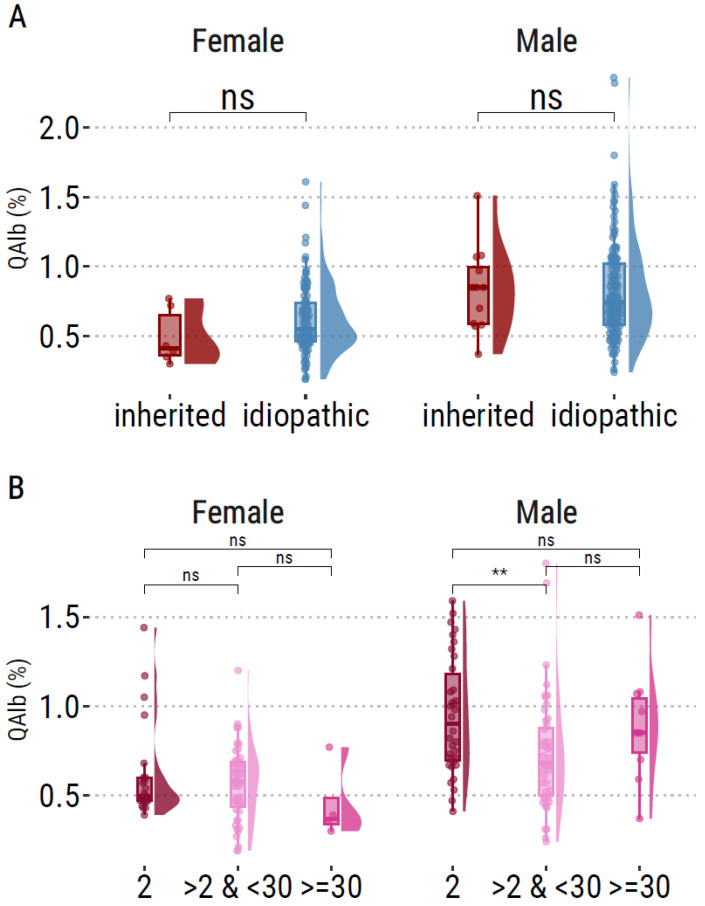
Evaluation of association between QAlb and genetic status. (**A**) Raincloud plots of QAlb between idiopathic and inherited forms of ALS, in males and females. (**B**) Raincloud plots of QAlb levels according to the number of hexanucleotide repeats in C9orf72 longest allele: 2, between 2 and 30, greater than or equal to 30. **: *p* < 0.01. ns: non significant.

**Figure 4 ijms-24-11196-f004:**
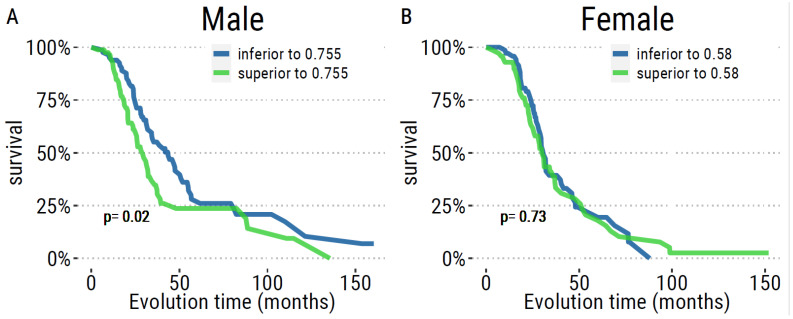
Kaplan–Meier curves with two groups set up according to the median value of QAlb (%) in male (**A**) and female (**B**) patients.

**Table 1 ijms-24-11196-t001:** Clinical characteristic of included patients.

Variable	All Patients (*n* = 307)	Male (*n* = 164)	Female (*n* = 143)	Adjusted *p*-Value
Age of onset (years)	66 (58, 73)	65 (56, 72)	67 (61.5, 74)	0.027
Site of onset				0.027
Spinal (N, %)	218 (71%)	128 (78%)	90 (62.9%)	
Bulbar (N, %)	89 (29%)	36 (22%)	53 (37.1%)	
Diagnostic delay (months)	9.0 (5.3, 14)	8.0 (5.0, 13.3)	8.9 (5.9, 14.0)	0.77
At diagnosis				
FVC (%)	95.5 (77.1, 111.2)	94.3 (78.6, 107.6)	100.7 (76.3, 114)	0.23
Weight (kg)	68 (59, 78)	74.5 (67, 81)	61 (54, 69)	<0.001
ALSFRS-r	40.5 (37, 43)	41 (37, 44)	40 (36, 42)	0.053
dALSFRS	0.30 (0.18, 0.55)	0.29 (0.17, 0.46)	0.35 (0.19, 0.59)	0.076
Variation in reference weight (%)	−1.52 (−7.69, 0.00)	−1.20 (−6.14, 0.00)	−2.63 (−8.90, 0.0)	0.19
Variation after a year (%)				
FVC	−18.6 (−32.3, −7.6)	−17.8 (−34.4, −7.4)	−18.7 (−30.4, −9.7)	0.93
Weight	−1.5 (−7.3, 2.5)	−1.9 (−7.0, 1.7)	−0.94 (−8.2, 4.0)	0.77
ALSFRS	−22.9 (−40.7, −12.5)	−21.7 (−39.4, −10.9)	−28.6 (−41.5, −14.3)	0.47
Disease duration before death (months)	26.3 (18.5–39.4)	25.8 (18–39)	28 (18.8–40)	0.77

Continuous variables are represented as median (interquartile range). Information about diagnostic delay, FVC at diagnosis, variation in ALSFRS, FVC and weight over a year, and about death was missing for 12.1, 19.9, 42.3 71, 45.6 and 33.3% of patients, respectively. ALSFRS-r: ALS functional rating scale-revised; dALSFRS-r: Progression rate of ALSFRS-R; FVC: Forced Vital Capacity.

**Table 2 ijms-24-11196-t002:** Multivariate Cox proportional-hazards model.

Variable	Male		Female	
	HR (95% CI)	*p*-Value	HR (95% CI)	*p*-Value
QAlb	2.3 (1.2–4.3)	0.009		
ALSFRS-r	0.95 (0.89–1.01)	0.099	0.94 (0.87–1.02)	0.14
dALSFRS-r	4.1 (1.4–11.6)	0.008	1.6 (0.45–5.8)	0.47
FVC	0.99 (0.98–1.0)	0.045	0.98 (0.97–1.0)	0.02
Variation of reference weight	1.0 (0.97–1.04)	0.81	0.98 (0.94–1.03)	0.46
Age of onset	1.0 (0.99–1.03)	0.57		
Onset site				
Spinal	0.53 (0.30–0.94)	0.031	0.64 (0.36–1.13)	0.12
Diagnostic delay	0.92 (0.88–0.97)	0.002	0.89 (0.84–0.95)	0.0002

HR: hazard ratio; CI: confidence interval. Variables with a *p*-value < 0.2 in the univariate models were included in a multivariate Cox proportional-hazards model after removal of highly correlated variables (r > 0.8).

## Data Availability

The datasets used and analysed during the current study are available from the corresponding author on reasonable request.

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
