# Peer review of "Implication of Central Nervous System Barrier Impairment in Amyotrophic Lateral Sclerosis: Gender-Related Difference in Patients"

_ijms, 2023, doi:10.3390/ijms241311196_

Round 1

Reviewer 1 Report

Thanks for the opportunity to review the paper titled “Implication of central nervous system barrier impairment in amyotrophic lateral sclerosis: gender-related difference in patients” by Alarcan H et al. The authors present an interesting topic to help better understand the pathophysiology of amyotrophic lateral sclerosis. The authors conclude that BBB/BSCB integrity is altered in many male ALS patients and that QAlb level is associated with survival in this subgroup but not in female. They did not report a specific association between BBB/BSCB alteration and inherited forms of the disease.

Overall, this is a well-written manuscript, and apparently the conclusions seems supported by the experimental data. Here are few of my suggestion that might help improve this manuscript:

1) In table 1: Please, indicate units of weight and FVC. Also, although FVC, ALSFRS-r and dALSFRS-r are defined on page 9, lines 276-7 and 284-5, it is recommended to define them in lines 75-7 as well, since the table must be self-comprehensive and furthermore is on page 2.

2) The conclusión of the study mentions that BBB/BSCB integrity is altered in many male ALS patients and that QAlb level is associated with survival in this subgroup but not in female, and this could be explained by a protective effect of oestrogen, but no hormonal measurements have been performed. This is an overstatement and the authors are recommended to provide evidence-based conclusion of the study.

3) Please, introduce DOI of reference 10.

4) References 16 and 17, change “Neurol. Sci. Off. J. Ital. Neurol. Soc. Ital. Soc. Clin. Neurophysiol” by “Neurol. Sci.

5) Reference 35 change “Amyotroph. Lateral Scler. Mot. Neuron Disord. Off. Publ. World Fed. Neurol. Res. Group Mot. Neuron Dis.” by “Amyotroph. Lateral Scler. Mot. Neuron Disord.

Reviewer 2 Report

Minor comments

Overall, the paper is well written. In general, this study shows a good comparision between male and female.

Line 16 & 82: It should be “Ninety- two” not “Nighty-two”

Figure 3A., statistical non significance shown is confusing. Both inherited and idiopathic are non-significant. Kindly indicate the statistically significant among the groups clearly, better to show with the comparison lines.

Major comments:

I do not have any major comments, however I doubt the hormones are not the only factor that make female less vulnerable to BBB and BSCB damage.

With respect to quantity of English, just few typographic errors need to be corrected.

Reviewer 3 Report

Reviewer comments and suggestions

The study discusses the involvement of central nervous system (CNS) barrier impairment in amyotrophic lateral sclerosis (ALS) condition by determining albumin quotient (QAlb) at the time of diagnosis of ALS in a large cohort of patients. 

The authors selected patients from the university hospital of Tours (n=307) who were included in this monocentric, retrospective study. Nighty-two patients (30%) had elevated QAlb levels. This percentage was higher in males (43%) than in females (15%). The study also analyzed a multivariate survival analysis and found that QAlb was significantly associated with survival in male patients (HR = 2.3, 95% CI = 1.2-4.3, p=0.009). Gender difference might guide the development of new drugs and help personalize the treatment of ALS.

Overall, the manuscript was well written. However, a few concerns/comments needed to be explained/modified. 

  1. Line 19 Full form of ALSFRS-r
  2. Line 36-39 I think one reference was not enough to describe all mechanism
  3. Line 58-59 Please discuss the study here
  4. Line 128-129 Here they discuss Figure 2 and figure 4 in the same paragraph
  5. Comments for Table 2 Why the authors did not mention the variables they adjusted in the legend part of the MS.
  6. Line 148-149 Please modify the line, as the information seems to be confusing.
  7. Line 154 a typo error was present, please check
  8. Line 232-233 what does the author means here, try to discuss either in a positive or negative so that it could be differentiated
  9. Line 269 The authors need to add ethical approval number and consent from the patients, it would be important to discuss in this section
  10. The authors need to add limitations to the study as well

Round 2

Reviewer 1 Report

The authors have satisfactory dealt with all previous concerns. The changes are especially appreciated.